# Factors Determining the Functional State of Cardiac Surgery Patients with Complicated Postoperative Period

**DOI:** 10.3390/ijerph19074329

**Published:** 2022-04-04

**Authors:** Alexey N. Sumin, Pavel A. Oleinik, Andrey V. Bezdenezhnykh, Natalia A. Bezdenezhnykh

**Affiliations:** Laboratory of Comorbidity in Cardiovascular Diseases, FSBSI Research Institute for Complex Issues of Cardiovascular Diseases, Sosnovy Blvd. 6, 650002 Kemerovo, Russia; pavel.oleinik.1991@mail.ru (P.A.O.); andrew22014@mail.ru (A.V.B.); n_bez@mail.ru (N.A.B.)

**Keywords:** functional status, muscle strength, cardiac rehabilitation, cardiac surgery, postoperative complications

## Abstract

The purpose of this work was to study the factors determining the functional state of cardiac surgery patients with a complicated postoperative period upon discharge from the hospital. This observational study included 60 patients who underwent cardiac surgery with a complicated postoperative course and with a prolonged intensive care unit stay of more than 72 h. We assessed handgrip and lower-extremity muscle strength and the six-minute walk test (6MWT) distance 3 days after the surgery and at discharge from the hospital. Some patients (53%) additionally underwent a course of neuromuscular electrostimulation (NMES). Two groups of patients were formed: first (6MWT distance at discharge of more than 300 m) and second groups (6MWT distance of 300 m or less). The patients of the second group had less lower-extremity muscle strength and handgrip strength on the third postoperative day, a longer aortic clamping time and a longer stay in the intensive care unit. Independent predictors of decreased exercise tolerance at discharge were body mass index, foot extensor strength and baseline 6MWT distance in the general group, duration of cardiopulmonary bypass in the NMES group and in the general group, and age in the NMES group. Thus, the muscle status on the third postoperative day was one of the independent factors associated with the 6MWT distance at discharge in the general group, but not in patients who received NMES. It is advisable to use these results in patients with complications after cardiac surgery with the use of NMES rehabilitation.

## 1. Background

The aim of cardiac surgery is to improve the prognosis and quality of life of cardiac patients. An improvement in quality of life is achieved by decreasing the severity of disease symptoms and increasing the patient’s functionality. However, the combination of surgery complexity and the use of cardiopulmonary bypass often leads to complications that require a longer stay in the intensive care unit and, accordingly, longer bed rest [1,2]. Harmful effects of immobility include decreased protein synthesis, increased proteolysis and loss of muscle mass and strength, which compromises functionality and the ability to carry out daily activities [3,4,5]. The effects caused by prolonged bed rest can last for months or even years after discharge from the intensive care unit [6,7], preventing patients from fully returning to baseline functionality [8], increasing the risk of re-hospitalization and greatly impairing daily activity and quality of life. However, adverse effects are not limited to the aforementioned parameters. Physical inactivity late after heart valve surgery is associated with increased long-term mortality [9]. It is clear that early mobilization of patients after surgery should be considered one of the therapy methods in these categories of patients. However, recent meta-analyses have obtained conflicting data: early mobilization improved exercise tolerance in patients after cardiac surgery in the work by Kanejima Y et al. [10] but had no effect on the length of stay in the intensive care unit or the total length of hospital stay in a study by Chen and colleagues [11]. This indicates the continuing relevance of research in this area [12,13], especially in the case of difficulties with the implementation of generally accepted rehabilitation programs. Accordingly, the purpose of this work was to study factors associated with the distance reached during the six-minute walk test by cardiac surgery patients with a complicated postoperative period upon discharge from the hospital.

## 2. Materials and Methods

### 2.1. Patient Population

In this observational study, we studied patients in the cardiac surgery hospital of the Research Institute for Complex Issues of Cardiovascular Diseases, hospitalized in the period from March 2017 to June 2019. Inclusion criteria: patients who underwent elective surgery on the heart or large intrathoracic vessels with a complicated early postoperative period. Complicated postoperative period was defined as the occurrence of any postoperative complication that increased the length of stay in the intensive care unit for at least three days or prolongation of mechanical ventilation. Exclusion criteria: patient refusal; severe comorbidity (neurological or orthopedic diseases) or cognitive dysfunction that interferes with the 6 min walk test and standard rehabilitation programs; postoperative delirium; agonizing patient; fatal postoperative complication resulting in hospital death. The study design is shown in Figure 1.

In total, from 1 June 2017 to 30 June 2019, the clinic performed 2171 operations on the heart and arterial vessels. Complications of the postoperative period were identified in 494 patients. Of these, 42 patients (1.9%) died in hospital due to fatal postoperative complications. Of the remaining 452 patients with a complicated postoperative period, 203 were assessed for preliminary compliance with the inclusion criteria. Of these, 109 patients were preliminarily approached according to the inclusion criteria. When talking with patients, 27 of them did not give their consent to participate in the study, 13 revealed additional exclusion criteria during the conversation, and 9 patients withdrew their informed consent at different stages of the study and did not complete all research procedures. Thus, the study included 60 patients (43 men, 17 women) from 52 to 70 years old who underwent cardiac surgery (coronary artery bypass grafting, valvular disease correction, heart transplantation and others, Table 1) and experienced postoperative complications. 

During their stay in the intensive care unit (ICU), patients participate in kinesiological exercises (massage, active and passive ventilation exercises, placement of patients in orthopedic chairs, breathing exercises). To assess the initial muscle status, all patients, starting from the third day of their stay in the intensive care unit, underwent handgrip and isokinetic dynamometry of lower-extremity muscles at the earliest opportunity. After transfer to the cardiac surgery department (on average, 5–6 days after the operation), patients were allowed to sit down, stand up, walk around the ward and perform therapeutic exercises. The six-minute walk test (6MWT) was also carried out at this time; the patients were mobilized before the test: they were allowed to walk around the ward and along the corridor. Then, walks along the corridor were added to the rehabilitation program, with a gradual expansion of the regimen under the supervision of a physical therapist. Instrumental examination included pre- and postoperative electrocardiography, echocardiography, and color duplex scanning of the brachiocephalic arteries and arteries of the lower extremities. Laboratory methods included the routine assessment of parameters in a fasting blood sample: glucose, creatinine and calculation of the glomerular filtration rate (GFR) using the CKD-EPI formula.

All patients underwent the usual program of inpatient postoperative rehabilitation. Some of the patients additionally received a course of neuromuscular electrostimulation (NMES). Dynamics of muscle status and 6MWT were evaluated 12–14 days after the initial assessment, on the eve of discharge from the cardiac surgery hospital.

According to the results of a six-minute walk test before discharge from the hospital, 2 groups of patients were formed (the median was used, and groups were evenly distributed): group 1 (6MWT > 300 m, *n* = 31) and group 2 (6MWT ≤ 300 m, *n* = 29). The groups were compared according to the data of preoperative and perioperative indicators, as well as data from functional examination in the early stages after the surgery. In addition, the factors associated with the 6MWT distance at discharge were assessed. The study protocol was approved by the local ethics committee. All patients provided informed consent after 72 h in the ICU to participate in the study. 

### 2.2. Assessment of the Functional Status

Dynamometry of the lower-extremity muscles was carried out using a manual isokinetic dynamometer “Lafayette MMT 01165” (USA) with the possibility of evaluating the obtained results directly on the screen of the device while taking the maximum force of contraction into account. After warming up the muscles, four exercises were performed in pairs on different muscle groups; the strength of the quadriceps femoris muscle, knee flexors, foot flexors and extensors was assessed. Handgrip strength was assessed using a dynamometer “DK-100” (Russian Federation); after warming up the muscles of the upper extremities, paired measurements of the right and left handgrip were carried out. The six-minute walk test was carried out in a closed room (a corridor with a pre-marked marking, with a flat floor, without an inclination in the horizontal plane) with the assessment of heart rate and blood pressure before and after the test, measuring the distance traveled.

### 2.3. Neuromuscular Electrostimulation 

The NMES course of the quadriceps femoris muscle was performed using the four-channel device “Beurer EM80” (Germany) according to the previously described method [14]. Each session was at least 90 min long, including 5 min warm-up and cool-down periods. Electrical stimulation was performed the day after dynamometry in the ICU. The duration of the NMES course was at least 7 sessions (12–14 on average) daily up to the day of discharge, applied during the entire period of the patient’s stay in the hospital in the postoperative stage. 

### 2.4. Statistical Analysis

Statistical processing of the obtained data was carried out using STATISTICA 10.0 (Dell Software, Inc., Round Rock, TX, USA) and SPSS-17 programs. The Shapiro–Wilk test was used to check the normal distribution, but since the distribution for all quantitative traits differed from normal, they are presented as medians and quartiles. We used the Wilcoxon test to assess intragroup differences and the dynamics of muscle status indicators, and the Mann–Whitney test was used for paired comparisons of intergroup differences. Nominal and binary characteristics were compared using the χ2 (chi-square) test with Fisher exact test for small samples. Stepwise multiple linear regression analysis was used to assess the relationship between the 6MWT distance at discharge and perioperative parameters, including maximal muscle strength. 

## 3. Results

The patients included in the study were predominantly male (72%) and had a median age of 63 [58; 68] and a frequent history of major cardiovascular events (46% had a history of myocardial infarction, and 13% had a stroke), as well as comorbidities such as chronic obstructive pulmonary disease (33%), diabetes mellitus (17%) and chronic lower-extremity ischemia (15%). The kidney filtration function, however, was typically preserved. The patients included in the study underwent a broad spectrum of surgeries. The overwhelming majority of operations were performed using cardiopulmonary bypass (90%), and the median duration of surgery was 4 h.

There was no difference between groups with different 6MWT distances on discharge in terms of preoperative parameters (Table 1) or types of surgical treatment, but the duration of cardiopulmonary bypass (CPB) and the aortic clamping time were higher in the group with a reduced 6MWT distance on discharge. Postoperative heart failure was more prevalent, and the intensive care unit stay was longer in patients in that group as well. The usual stay of a patient in the hospital after cardiac surgery in our clinic is about 7 days. If necessary, treatment can be extended at the stage of inpatient rehabilitation (this was exactly the case with our patients), and hospital length of stay did not differ in groups.

When assessing the muscle status on the third day after surgery, the patients of the group with a reduced 6MWT distance had decreased strength in almost all muscle groups studied, except for the quadriceps muscles (Table 2, Figure 2 and Figure 3). These patients had a significantly longer stay in the intensive care unit and poorer results of 6MWT (Table 1 and Table 2, Figure 4).

When the test was repeated, the muscle strength increased significantly, while at the time of discharge, the differences in the strength of the muscle groups remained (Table 3). The frequency of NMES during inpatient rehabilitation did not differ between the groups either.

Multiple linear regression was used to assess the association of 6MWT distance at discharge (dependent variable) with perioperative parameters. The baseline model included the following variables: age, body mass index, CPB time, aortic clamping time, GFR, maximum strength of all muscles, baseline 6MWT distance (Table 4). After calculating the BSA, adding this variable did not change the results of the linear regression (B = 1.795; Beta = 0.198; *p* = 0.246), so we did not change our original model. In the entire sample, the distance of the initial 6MWT, the duration of cardiopulmonary bypass, the body mass index and the strength of the left foot extensor statistically significantly predicted the 6MWT distance at discharge (F (4, 61) = 18.852, *p* < 0.0001, R2 = 0.774 (Appendix A)). In the NMES group, only the age and duration of cardiopulmonary bypass significantly predicted the 6MWT distance at discharge (Table 4), F (2.61) = 12.478, *p* = 0.002, R2 = 0.714 (Appendix A). In the patients without NMES, the following factors were statistically significant for the 6MWT distance at discharge: body mass index, time of baseline 6MWT and strength of the left foot extensor (Table 4), F (3.61) = 20.780 *p* < 0.0001, R2 = 0.862 (Appendix A).

## 4. Discussion

In our study, patients with complications after cardiac surgery, a highly selected population, showed a decrease in 6MWT distance at discharge. Preoperative factors (age, body mass index), perioperative factors (CPB duration), muscle status on the third postoperative day and functional state (in patients without NMES) after their stay in the intensive care unit were found to be independent factors associated with the 6MWT distance at discharge. 

These findings are consistent with the national registry of Korea, which showed that 49% of patients were physically inactive after heart valve surgery [9], with 39% only becoming inactive after surgery. Japanese researchers found that 18% of patients undergoing cardiovascular surgery experienced a decrease in the activity of daily living (ADL) score at discharge [12]. Three months after surgery, a functional decrease in ADL was observed in 45% of patients aged 65–79 years and in 56% aged 80 years and older. One year after surgery, functional outcomes continued to decrease in 38% and 57% of cases, respectively [15]. According to our data, similar findings were obtained, although we studied only patients with complications after cardiac surgery. 

When assessing preoperative factors determining postoperative functional status in older age groups, it was shown that low baseline gait speed and cognitive impairment were predictors of a decrease in ADL after surgery [12]. In a study by Radi B. et al., the distance of 6MWT at discharge after CABG was 314 ± 76 m, and that after valvular surgery was 328 ± 71 m. The 6MWT distance was significantly associated with gender, age, diabetic status, height and presence of atrial fibrillation in patients after CABG and with gender, age and atrial fibrillation in patients undergoing valve surgery [16]. When perioperative factors were taken into account, the 6MWT distance at discharge in patients undergoing cardiac surgery was associated with the type of surgery, CPB time, functional independence measure (FIM) scores and body mass index [17]. This is quite consistent with our results; we assessed the muscle status on the third day after surgery and before discharge instead of FIM, and decreased muscle strength was found in the group with low functional status. 

The determination of preoperative factors associated with a low functional state after surgery allows high-risk patients to be identified even before surgery, which makes it possible to reconsider treatment strategies. Additionally, in such cases, it is possible to carry out pre-rehabilitation [18,19] or to use early patient mobilization [10,20]. It has been convincingly demonstrated that early patient mobilization can improve the physical capabilities of patients by the time of discharge from the hospital [10]. However, firstly, the search continues for the optimal rehabilitation program (intensity, type, frequency and duration of exercise) [20], and secondly, early mobilization becomes problematic with a complicated postoperative period. For example, early mobilization of cardiac surgery patients has been associated with a number of side effects, including significant hemodynamic changes, which should be closely monitored in the intensive care unit [21]. It may also require additional staffing with adequate training in the rehabilitation of these patients [22]. In addition, postoperative in-hospital mobilization should be adjusted to the patient’s functional status at that specific moment, building up from sitting in a chair (which should be initiated as early as possible, including in the ICU) and increasing the functional status by walking, biking on an exercise bike or even walking on stairs [23,24]. Otherwise, early mobilization may not lead to a reduction in hospitalization times or a reduction in the cost of treatment [11].

In our work, we studied patients with a complicated postoperative period—that is, patients in whom the greatest decrease in functional capabilities can be expected with further observation. Nevertheless, the spread in the values of the 6MWT distance at discharge turned out to be quite pronounced. In addition, we received, to some extent, an answer to the question: On what factors does this parameter depend? It became clear that muscle status is one of these factors, and therefore, interventions targeting skeletal muscles appear to be quite reasonable. Yes, the NMES by itself was unable to affect the 6MWT distance, which can be indirectly judged by the results of multiple regression analysis in this work, as well as in a randomized pilot study published earlier [14]. However, in the NMES group, the 6MWT distance was no longer associated with the strength of any muscle group, and the strength of the stimulated quadriceps muscle did not differ in the groups with high and low load tolerance at discharge. Therefore, NMES can be considered as one of the options for rehabilitation. Apparently, this type of passive muscle exercise should be started in the intensive care unit as early as possible (we were limited by the design of the study, where such exercises began after the initial test on the third postoperative day) and include not only the most significant muscle groups (such as the quadriceps) but also other muscles of the lower extremities (for example, the calf muscles), which had the strongest association with the 6MWT distance at discharge. The undoubted advantage of this type of training, in contrast to traditional approaches [25], is the possibility of starting it at the earliest possible time in sedated patients and those on mechanical ventilation [26]. Without a doubt, passive muscle exercises can be performed not only with NMES but also by a physical therapist. Comparing these two types of passive muscle training should be a topic for future research.

One feature of the present study was the low values of the 6MWT distance at discharge. For instance, in the study by Hirschhorn et al. [27], at discharge from the hospital after CABG in the training group (walking), the average distance of the 6MWT was 444 m, and even in the control group, it was 377 m, which is significantly higher than in the present study. These differences are explained by the fact that the aforementioned study included patients with an uncomplicated postoperative period (postoperative ventilation time was 10,4 h on average); therefore, their muscle status and functional state decreased to a lesser extent. It is also difficult to interpret the association of the 6MW distance with the muscle strength of the left lower limb. Perhaps this is due, to some extent, to the local location of the saphenous vein for coronary bypass grafting.

## 5. Study Limitations

First, this study was carried out in only one center, which limits the broader applicability of the data obtained. Second, the dividing criterion of a 6MWT distance of more or less than 300 m was chosen quite arbitrarily, based on the data that a distance < 300 m in the 6MWT group is a predictive marker of mortality and readmission [28] and also considering the data of a meta-analysis on early mobilization after cardiac surgery, in which the 6MWT in the control groups at discharge ranged from 272 to 331 m [10]. Third, in our study, we assessed the functional state of patients after surgery (muscle status on day 3, 6MWT after transfer to the somatic department), which did not allow us to compare the dynamics of muscle and functional status to the preoperative state. This was due to the design of the study, which only included patients with a complicated postoperative period. Fourth, some of the patients received an additional course of NMES, which could affect the relationships that we studied. However, in a previous study [14], we did not reveal the effect of this rehabilitation method on the 6MW distance based on the median; therefore, we considered it possible to analyze the data in one cohort.

## 6. Conclusions

Patients with complications after heart surgery had low 6MWT values at discharge. The patients of a group with a lower functional state had decreased lower-extremity muscle (knee flexors, foot flexors and extensors) strength and handgrip strength on the third postoperative day, a longer aortic clamping time and CPB duration, and a longer stay in the intensive care unit. Independent predictors of decreased exercise tolerance were body mass index, left foot extensor strength and baseline 6MWT distance among patients without NMES and in the sample as a whole; the duration of CPB in the NMES group and in the general group; and age in the NMES group. It is possible to use these results in patients with complications after cardiac surgery with the use of NMES rehabilitation. However, the optimal NMES technique (time of onset after surgery, stimulated muscle groups) requires further research. 

## Figures and Tables

**Figure 1 ijerph-19-04329-f001:**
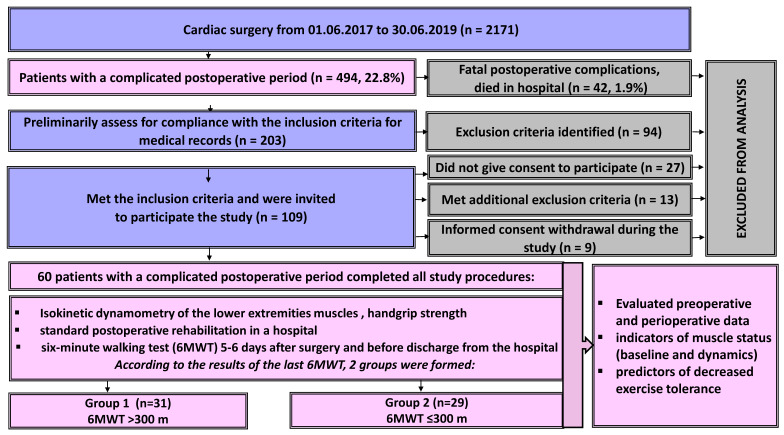
Study design. Patient selection and inclusion flowchart.

**Figure 2 ijerph-19-04329-f002:**
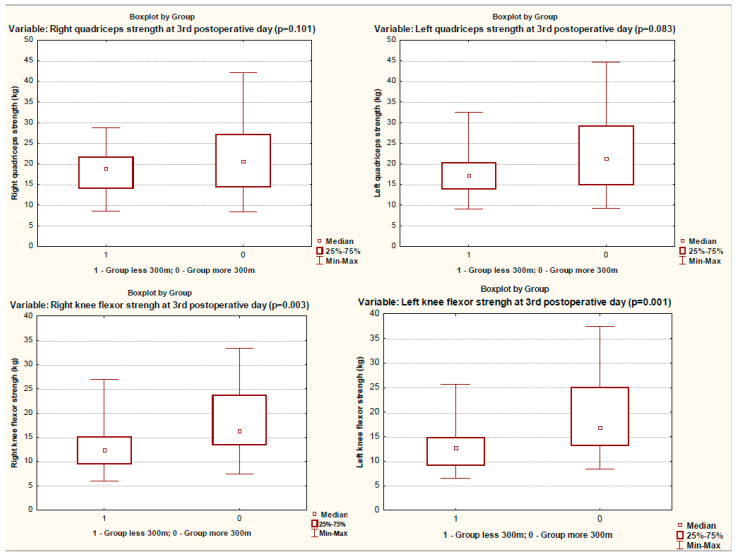
Quadriceps and knee flexor strength in groups on 3rd postoperative day.

**Figure 3 ijerph-19-04329-f003:**
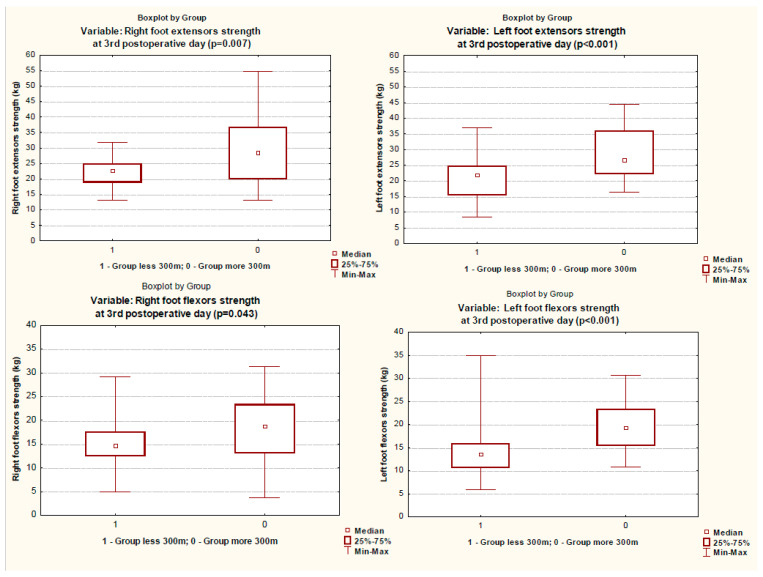
Foot extensor and flexor strength in groups on 3rd postoperative day.

**Figure 4 ijerph-19-04329-f004:**
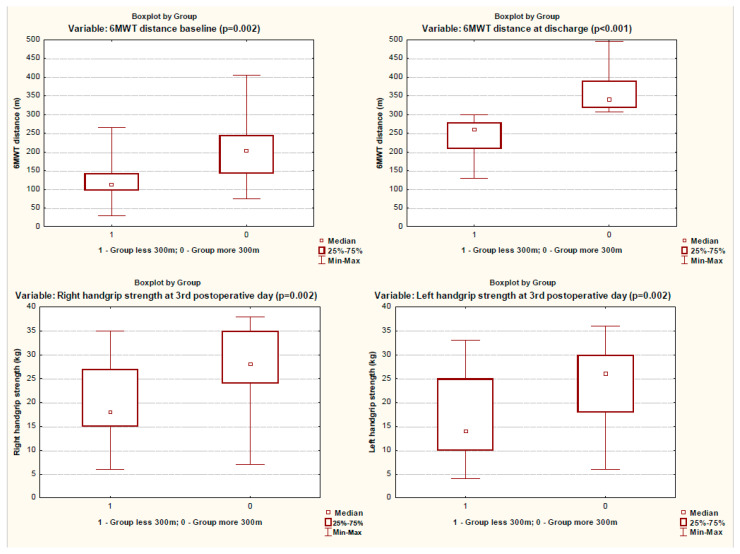
The 6MWT distance at baseline and at discharge and handgrip strength on 3rd postoperative day in groups.

**Table 1 ijerph-19-04329-t001:** Baseline characteristics of patients in groups.

	Group 1 (*n* = 31)	Group 2 (*n* = 29)	*p*
Men (n, %)	26 (83.9)	17 (58.6)	0.045
Age (years)	62.0 [58.0; 65.0]	65.0 [58.0; 70.0]	0.131
Height (cm)	172.0 [167.0; 176.0]	167,0 [158.0; 174.0]	0.020
Weight (kg)	82.0 [70.0; 95.0]	75.0 [65.0; 85.0]	0.160
BMI (kg/m^2^)	28.4 [24.2; 31.9]	27.7 [24.5; 30.7]	0.971
BSA (m^2^)	1.94 [1.79; 2.05]	1.86 [1.73; 1.92]	0.039
GFR (mL/min/1.73 m^2^)	83.4 [76.3; 91.9]	82.4 [75.7; 89.7]	0.390
NYHA class ≥ 3 (n, %)	6 (19.4)	10 (34.5)	0.247
MI history (n, %)	15 (48.4)	13 (44.83)	0.802
PCI history (n, %)	6 (19.4)	8 (27.6)	0.547
Hypertension (n, %)	25 (80.7)	27 (93.1)	0.257
Stroke history (n, %)	2 (6.5)	6 (20.7)	0.140
Permanent AF (n, %)	8 (25.8)	10 (35.7)	0.576
Diabetes mellitus (n, %)	5 (16.1)	5 (17.9)	0.590
COPD (n, %)	11 (35.5)	9 (31.0)	0.788
CLI ≥ 2A history (n, %)	2 (6.5)	3 (10.3)	0.666
ICA stenosis > 50%	4 (14.8)	6 (21.4)	0.500
NMES (n, %)	14 (45.2)	18 (62.1)	0.208
Preoperative Examination Data			
End-diastolic volume (mL)	194.0 [141.0; 272.0]	187.0 [130.0; 247.0]	0.976
End-systolic volume (mL)	83.0 [47.0; 141.0]	79.0 [54.0; 141.0]	0.420
Left ventricular ejection fraction (%)	60.0 [39.0; 65.0]	57.0 [38.0; 64.0]	0.762
SPAP, mm Hg	32.0 [24.5; 37.0]	39.0 [29.0; 51.0]	0.051
Types of Surgical Interventions and Their Characteristics			
Isolated CABG (n, %)	7 (22.6)	5 (17.2)	0.750
Aortic valve replacement (n, %)	1 (3.2)	3 (10.3)	0.346
Mitral valve replacement (n, %)	5 (16.1)	1 (3.5)	0.196
CABG+carotid endarterectomy	2 (6.5)	0	0.492
Combined CABG and valve replacement (n, %)	3 (9.7)	7 (20.7)	0.175
Multivalve operations (n, %)	3 (9.7)	2 (6.9)	0.532
Elective thoracic aorta surgery (n, %)	2 (6.5)	4 (13.8)	0.417
Aortic dissection (n, %)	1 (3.2)	2 (6.9)	0.606
Heart transplantation (n, %)	2 (6.5)	2 (6.9)	0.668
CABG+ventriculoplasty/thrombectomy/radiofrequency ablation	5 (16.1)	3 (10.3)	0.708
Cardiopulmonary bypass (n, %)	28 (90.3)	26 (89.7)	0.632
Cardiopulmonary bypass duration (min)	109.5 [72.5; 158.5]	145.5 [117.0; 194.0]	0.007
Aortic clamping time (min)	75.0 [45.0; 104.0]	104.0 [81.0; 130.0]	0.007
Total duration of surgery (min)	210.0 [190.0; 260.0]	245.0 [160.0; 370.0]	0.384
Peri- and Postoperative Complications in Groups			
Myocardial infarction (n, %)	0 (0)	2 (6.9)	0.229
Heart failure (n, %)	4 (12.9)	12 (41.4)	0.019
Atrial fibrillation (n, %)	10 (32.3)	13 (44.8)	0.427
Sternal wound complication (n,%)	4 (12.9)	5 (17.2)	0.727
Multiple organ failure (n, %)	5 (16.1)	10 (34.5)	0.139
Respiratory failure (n, %)	5 (16.1)	7 (24.1)	0.527
Pericardiocentesis (n, %)	3 (9.7)	1 (3.5)	0.613
Acute renal failure with a course of renal replacement therapy (n, %)	2 (6.5)	7 (24.1)	0.076
Intensive care unit length of stay (d)	5.5 [3.0; 6.0]	7.5 [3.0; 12.0]	<0.001
Hospital length of stay (d)	28.5 [21.0; 36.0]	33.5 [26.0; 42.0]	0.089

Notes: BMI—body mass index; CLI—chronic limb ischemia; COPD—chronic obstructive pulmonary disease; BSA—body surface area (Du Bois method); GFR—glomerular filtration rate; PCI—percutaneous coronary intervention; NYHA—New York Heart Association; CABG—coronary artery bypass grafting; SPAP—systolic pulmonary artery pressure; all qualitative features are presented as numbers and percentages, and all quantitative ones are presented as the median and upper and lower quartiles.

**Table 2 ijerph-19-04329-t002:** Muscle status on postoperative day 3 in groups.

	Group 1 (*n* = 31)	Group 2 (*n* = 29)	*p*
Right quadriceps strength (kg)	20.5 [14.4; 27.2]	18.9 [14.0; 21.8]	0.101
Left quadriceps strength (kg)	21.2 [14.9; 29.3]	17.1 [13.8; 20.3]	0.083
Right knee flexor strength (kg)	16.3 [13.4; 23.8]	12.3 [9.5; 15.1]	0.003
Left knee flexor strength (kg)	16.7 [13.2; 25.1]	12.6 [9.1; 14.9]	0.001
Right foot extensor strength (kg)	28.4 [20.0; 36.8]	22.5 [18.9; 24.9]	0.007
Right foot flexor strength (kg)	18.7 [13.1; 23.4]	14.6 [12.5; 17.6]	0.043
Left foot extensor strength (kg)	26.7 [22.2; 36.1]	21.7 [15.4; 24.9]	<0.001
Left foot flexor strength (kg)	19.2 [15.4; 23.3]	13.6 [10.7; 15.9]	<0.001
Distance of 6MWT (m)	202.5 [142.0; 245.0]	112.0 [97.0; 143.0]	0.002
Right handgrip strength (kg)	28.0 [24.0; 35.0]	18.0 [15.0; 27.0]	0.002
Left handgrip strength (kg)	26.0 [18.0; 30.0]	14.0 [10.0; 25.0]	0.002

**Table 3 ijerph-19-04329-t003:** Muscle status at discharge in groups.

	Group 1 (*n* = 31)	Group 2 (*n* = 29)	*p*
Right quadriceps strength (kg)	26.2 [21.4; 29.6]	23.3 [18.0; 27.7]	0.083
Left quadriceps strength (kg)	25.9 [20.7; 30.0]	22.3 [18.1; 26.1]	0.056
Right knee flexor strength (kg)	21.9 [15.8; 25.7]	14.2 [11.8; 20.1]	0.003
Left knee flexor strength (kg)	21.8 [16.9; 27.7]	15.7 [11.3; 20.2]	0.001
Right foot extensor strength (kg)	33.9 [25.2; 38.9]	25.4 [22.9; 27.8]	<0.001
Right foot flexor strength (kg)	20.4 [18.0; 25.6]	16.2 [13.9; 19.2]	<0.002
Left foot extensor strength (kg)	31.7 [24.7; 37.2]	23.2 [19.0; 27.7]	<0.001
Left foot flexor strength (kg)	20.8 [17.8; 24.0]	14.9 [12.9; 17.4]	<0.001
Distance of 6MWT (m)	341.0 [318.0; 390.0]	259.0 [208.0; 278.0]	<0.001
Right handgrip strength (kg)	32.0 [26.0; 37.0]	22.0 [18.0; 31.0]	0.003
Left handgrip strength (kg)	28.0 [22.0; 32.0]	18.0 [15.0; 30.0]	0.002

**Table 4 ijerph-19-04329-t004:** Linear regression analysis (stepwise method) for the relationship of distance 6MWT at discharge with other variables (*n* = 60).

Dependent Variable: Distance 6MWT	Unstandardized Coefficients		Standardized Coefficients	t	Sig.
	B	Std. Error	Beta		
Model 1 (total)					
(Constant)	452.971	72.222		6.272	0.000
6MWT after intensive care unit	0.273	0.151	0.234	1.803	0.085
Cardiopulmonary bypass duration	−0.444	0.124	−0.430	−3.564	0.002
Body mass index	−6.691	1.712	−0.401	−3.909	0.001
Left foot extensor strength	3.114	1.235	0.308	2.522	0.019
Model 2 (patients with NMES)					
(Constant)	622.851	70.079		8.888	0.000
Age	−4.095	1.153	−0.624	−3.552	0.005
Cardiopulmonary bypass duration	−0.380	0.156	−0.427	−2.430	0.035
Model 3 (patients without NMES)					
(Constant)	484.451	81.021		5.979	0.000
Body mass index	−9.315	1.960	−0.588	−4.753	0.001
6MWT after intensive care unit	0.332	0.140	0.327	2.376	0.039
Left foot extensor strength	3.162	1.388	0.321	2.278	0.046

## Data Availability

Data regarding this manuscript are available in the Federal State Budgetary Scientific Institution “Research Institute for Complex Issues of Cardiovascular Disease”, Kemerovo, Russia.

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
