# Peer review of "Factors Determining the Functional State of Cardiac Surgery Patients with Complicated Postoperative Period"

_ijerph, 2022, doi:10.3390/ijerph19074329_

Round 1

Reviewer 1 Report

The topic of the article is important - regarding planning the volume of surgical / therapeutic procedure for heart diseases patients, regarding early ambulation and rehabilitation programs, prognosis of potential patients' quality of life post cardiac surgery. The methods suggested in the manuscript look useful, reproducible  and helpful. The material and methods are well described, and the results are clearly shown with tables and figures. More than half of the citations are published during the last 5 years. 

The discussion and conclusions are useful for the readers. 

Author Response

Dear colleague!
Thank you for appreciating our article. This inspires us to continue our work in this direction. 

Reviewer 2 Report

Dear Editor and Authors,

It was a pleasure to evaluate the manuscript "Factors determining the functional state of cardiac surgery patients with complicated postoperative period" for publication in IJERPH. Thank you for your invitation.

The aim of this study was to was to study factors determining the functional state of complicated cardiac surgery patients with prolonged ICU stay upon hospital discharge. The authors found that 6-minute walk test distance (6MWT) of patients with a 6MWT distance >300m at discharge, were more likely to have a lower 6MWT earlier after surgery, compared to patients with a 6MWT distance <300m, among other factors.

The subject addressed in this article is worthy of investigation, and the conclusions are (partially) supported by the data collected. There are serious reservations regarding novelty of the work, methodology, and interpretations of the results.

I would like to comment as follows:

  1. There is tremendous overlap in the study protocol and results from a RCT from the same group in the same period and patient group on the effectiveness of neuromuscular electrical stimulation (NMES) in early rehabilitation of patients with postoperative complications after cardiovascular surgery, published in 2020 (doi: 10.1097/MD.0000000000022769). It is unclear how the underlying study differs from the RCT, if the same patients were analysed in both studies, and how conclusions from both studies differ.
  2. IJERPH focuses on public health, yet this study focuses on a very specific population. Is this study more relevant for MDPI Healthcare, especially the special issue on Physiotherapy and Cardiothoracic Care in Acute and Chronic Care?
  3. The purpose/aim on lines 52-54 is vague. Be more specific, and use for instance the PICO framework to structure the aim.
  4. The design of the study is not clear. Use the STROBE checklist (and provide us with it in your revised manuscript) for observational studies, and also include primary and secondary endpoints
  5. Patient population consisted of complicated cardiac surgery patients aged 52-70 years. There seems to be selection bias, why was no patient older than 70 years, as the average age for cardiac surgery (in general) is 68 years?
  6. Why were the groups divided in >300m or <300m at 6MWT before discharge from the hospital? Is there scientific evidence for this cut-off, or was the median used, and groups evenly distributed? How does 300m compare to previous studies?
  7. Lines 100-101: Please confirm that patients gave informed consent at the ICU after 72 hours to participate in the study
  8. A Chi2 test with Yates correction was used and not a Fisher Exact test. The Yates correction is not as reliable. Please recalculate with Fisher exact test.
  9. Results, Table 1: There seems to be a difference between groups regarding sex distribution. Group 1 <300m 84% male and Group 2 59% male. Both with chi2 and Fisher exact, this p-value is < 0.05. Please explain why 0.06 was showed here in the table.
  10. Table 1: A difference in height can be explained by the unequal sex distribuation. Please also calculate the BSA (Du Bois), as this can be important for comparison. Additional reading: https://doi.org/10.3390/s21061979
  11. Table 1: the surgical interventions consist of a complex group of patients. With the selection criteria (ICU > 72 hours), the population can be representative. Also include percentage off-pump surgery in both groups. Confirm that listed complications are perioperative complications, or are they postoperative? Define acute renal failure (i.e. AKIN categories). Remove red formatting of some p < 0.05 values.
  12. Figures 2-4: Name group 1 and 2, just as in the text instead of 0 and 1. Define box-and whiskers, are they according to Tukey? Align vertical axis from 0 to 50 for every graph, otherwise comparisons can not be made visually. Furthermore, incorporate the data from 3rd postoperative day and before discharge in one graph, as grouped data (so 4 box-and whiskers in one panel)
  13. Update the regression model with variable sex, and if a difference was found, also BSA. Why was NMES not a variable?
  14. Lines 181-182: Supplemental tables 1 and 2 were not shared with the reviewers.
  15. Table 5: Why were 61 included in the linear regression analysis, but 60 (29+31) in all others?
  16. Discussion: Start with stating your study population group, as this is a highly selected population.
  17. Lines 191-194 "In our study, approximately half of the patients with complications after cardiac surgery showed a decrease in functional status (6MWT distance)." How is this statement supported by your study results? Almost all patients in this study have a decrease in functional status: early mobilization strategies increased 6-minute walking assessment distance from 377m to 444m doi:10.1016/j.hlc.2007.09.004.
  18. Line 194: what is the "functional state"? 6MWT? As this was not significant in a linear regression analysis (p = 0.085). Please explain.
  19. Line 206: low baseline gait speed and cognitive impairment can indeed be predictors. Why were patients not informed and measured before surgery, to see if baseline preoperative characteristics determined outcome? Patients without postoperative complications could then be excluded from postoperative measurements. This seems to be a limitation of this study
  20. Lines 218-222: This is a too popularly written section, and was not the topic of the study, that should be rewritten or omitted for scientific publishing purposes. Also, high-risk patients could benefit from preoperative optimalisation with respect to diet, mobilisation, hemoglobin optimisation, etc. Please comment.
  21. Lines 222-226: Indeed early patient activation is important to increase patient functional levels. See doi:10.3390/healthcare9121735. See also (https://doi.org/10.3990/1.9789036550697, Chapter 7, page 132-156) Note: The authors should not be compelled to cite this work, unless they feel that it strenghtens the paper.
  22. Lines 229-232 are interesting, but slightly limited in view. First, postoperative in-hospital mobilisation should be adjusted to patients functional status at that specific moment, and build up from sitting in a chair (which should be done as early as possible, also at ICU), and increase the functional status by walking, biking on an exercise bike, or even walking the stairs. See also references under comment 21.
  23. The study limitations section is an exact duplicate of the discussion section. Please change accordingly.
  24. Conclusion first sentence: "Among patients with a complicated postoperative period after cardiac surgery, a decrease in the 6MWT distance to 300 m or less at discharge was noted in 48.3% of cases." does not add value to the manuscript, as it is per definition of your group division (50% in each group), please comment.
  25. Data Availability Statement: "Data sharing not applicable". Explain why not. The informed consent form as disclosed under the non-published word document explicitly states: "5. I voluntarily agree that the results obtained during the survey, including data on my physical and mental health or condition, may be used for scientific purposes and published in scientific literature, subject to full confidentiality."

Minor comments:

  1. Remove affiliations 2-4 as they are identical, and consider making an authors biography with more specific information on function if required.
  2. Abstract: The abstract is unstructured without subheadings. Is this according to the author guidelines? Remove the NMES details from lines 24-27 as this is not part of the underlying study.
  3. It is unclear from the abstract that the patient population is about postoperative patients with a prolonged ICU stay of more than 72 hours (as it seems to be). Be more specific.
  4. Introduction: lines 44-45 refer to reference 7. That study is about late inactivity, measured with a generic questionnaire, and is not about early mobilisation such as described in the manuscript. Please adjust this in the manuscript.
  5. There is recent literature available on the positive effects of early mobilisation after cardiac surgery, please see https://doi.org/10.3390/
    healthcare9121735
  6. Materials and methods: line 60 states "large vessels". If these are the large intrathoracic arteries, please clarify.
  7. Lines 61-64 be more specific: Are patients included with an ICU stay > 72 hours, regardless of a complication? How long is prolonged ventilation time? which severe comorbidities were excluded? How does the standard rehabilitation program look like (preferably with activity milestones)
  8. Figure 1 is unclear: Consider to use a CONSORT flow diagram. Resolution is too low, consider exporting the figure as vector image.
  9. Line 74: Only 50% of the patients matching the inclusion criteria were approached. Please explain why.
  10. Line 84: Is it correct that patients stayed for at least 3 days at ICU, and 3 days at a surgical ward before the first 6MWT? What kind of mobilisation activities were performed before this first measurement?
  11. Lines 93-94: 12-14 days is an (extremely) long hospital stay, even for complicated patients. What is a normal postoperative stay in Russia? Please compare this to international references, see doi:10.1136/bmjopen-2017-016947, doi:10.1016/j.athoracsur.2021.03.043, doi:10.1161/CIRCOUTCOMES.116.003327.
  12. Throughout the manuscript, i.e. line 134: Remove excessive decimals for clarity: Values >10 without a decimal; Values 1-10 with 1 decimal; Values < 1 with 2 decimals
  13. Line 156: Is admission correct, or should this be discharge? From ICU or surgical ward?
  14. Table 2: Table title states: "Muscle status at postoperative day 3 in groups." Confirm that your complicated patients walked a 6MWT at the ICU on postoperative day 3?
  15. Line 166: include NMES frequency in Table 3
  16. Line 184 "control group" seems to be referring to the Authors' 2020 paper. Explain.
  17. Discussion, Line 194: Be more specific: only left foot extensor strenght was found associated with 6MWT at discharge. Did the authors investigate why left only? Did this correspond to the side of preoperative claudication?
  18. Lines 196-198 see earlier comment on reference 7. This is not relevant for this study, but could be important for the long term, which was not investigated in this study (consider refering to it as "future work plans")
  19. Lines 208-210: state the actual distances achieved in other studies
  20. Lines 246-251: Passive muscle exercise as early as possible is indeed very important, but should this be done with NMES, or by a physical therapist? This should be topic of future studies, please elaborate on your ideas for such a study.
  21. Excessive extra white spacing is found throughout the manuscript (i.e. line 322), please adjust this accordingly throughout the manuscript.
  22. Line 325 "foot extensor strength" should refer to left, but not right. Please make your conclusion paragraph consistent with results.

Reviewer 3 Report

Many thanks to the editor for giving me the opportunity to review this article. The objective of the article is to study the factors determining the functional state of cardiac surgery patients with complicated postoperative period upon the discharge from the hospital. The results presented by the authors are interesting. However, there are several major concerns that need to be addressed.

 General considerations

  • In the introduction there are some statements that are not supported by any references. The authors are advised to review this.
  • The method is correctly described.
  • The results are presented clearly and concisely.
  • The discussion, in general, repeats what is stated in the results. In my opinion, more should be done to try to explain, based on the existing evidence, the reasons for the results found.
  • The references section should be revised to fit the format of the journal.
  • The references provided are up to date and relevant.
  • English should be revised throughout the manuscript.

Reviewer 4 Report

Dear Authors,

Thank you for your work in this important and interesting topic, but I believe some major revisions will be required. Here are my questions/concerns:

  1. I am not entirely sure what research question you are trying to answer. You should be more clear. The statistical analysis is a comparison of high and low 6MWT times, but it is unclear to be if that is aligned with the purpose of the study. Why did you not compare complicated to uncomplicated patients?
  2. Is the study adequately powered? Was there a power analysis conducted? Since logistic regression was used in the analysis I am concerned that the sample might not have been adequate.
  3. What is the justification for making 300 ft. the cut point for the 6MWT?
  4. You say 53.3% of sample had a NMES. How was that determined and how what was the breakdown of that group?
  5. Also, you should be more clear about the NMES, its role, and how it was measured.
  6. Group 2 had significantly longer cardiopulmonary bypass duration and ICU duration. Could that not explain the differences between the groups, independent of their functional capacity?
  7. Logistic regression is a complicated analytical test. There should be more description about the use of logistic regression. Was a Hosmer-Lemeshow test conducted to determine the fitness? Was a pseudo r squared test used? What was the result of the model, and what is the implication?
  8. In the discussion, lines 218 and 219 state that it is possible to abandon major cardiac surgery in favor of minimally invasive or transcatheter or conservative therapy. I am not sure this statement can be justified since the indication for surgery is more complicated than having a low functional capacity.
  9. In the conclusion the authors state that it is advisable to use NMES in patients with complications. What evidence does that study have to support that statement?

Round 2

Reviewer 2 Report

Dear Editor and Authors,

It was a pleasure to see improvements (Revision 1) on the manuscript "Factors determining the functional state of cardiac surgery patients with complicated postoperative period" for publication in IJERPH.
The authors addressed most comments satisfactory, conducted additional and revised statistical analyses, updated figures, and all on a short notice. Overall, the manuscript improved. However, some comments were only answered in the point-by-point response, and not implemented in the manuscript. The authors' response to other reviewers, especially the extensive comments of Reviewer 4, are missing.

Comments on original submission

Major comments:

Comments 2, 3, 7-9, 14-16, 19-21, and 23 were answered to satisfaction, and improved the manuscript.

1. The authors elaborated now more on the difference between the original NMES RCT and the underlying observational study, thank you. a) To what extent do these studies overlap? b) Be explicit that patients in the original RCT were also included in the underlying study. c) Did patients give informed consent for reuse of their data for other studies than they gave consent to? Please comment and address accordingly.

I would like to quote the STROBE explanation on study design: "The secondary use of existing data is a creative part of observational research and does not necessarily make results less credible or less important. However, briefly restating the original aims might help readers understand the context of the research and possible limitations in the data." From: doi:10.1371/journal.pmed.0040297

4. The STROBE checklist is missing from the documents submitted, please add. Furthermore, some items are still not metnioned in the revised manuscript.

5. Thank you for providing references of previous studies from your group. Do heart team discussions in your centre favour CABG for younger patients, compared to PCI? I still find it curious that no patients aged >70 are in your study, as you would expect a more complicated stay in these older patients.

6. The median is used to divide groups; that can be a sound choice. Compared to other studies than already referenced, the 6MWT seems low. 
Please include studies that are not in line with your findings in your discussion as well. e.g. doi:10.1016/j.hlc.2007.09.004. Furthermore, what is the minimal clinically important difference in the six-minute walk test? See doi:10.1081/copd-200050527. Please comment.

10. Thank you for adding BSA to Table 1 as a factor. Also include BSA abbreviation incl the calculation method (Du Bois) in Table 1 notes

11. Your explanation is clear, yet these changes were not implemented in the revised version of the manuscript. a) include this in table 1 under isolated CABG and/or add CPB use under inclusion criteria; b) add renal replacement rherapy to the AKI line in Table 1; c) Change perioperatieve to peri- and posteropative compl in Table 1

12. This comment was partially addressed. Comment and/or change the figures on these 2 comments: "Align vertical axis from 0 to 50 for every graph, otherwise comparisons can not be made visually." and "Furthermore, incorporate the data from 3rd postoperative day and before discharge in one graph, as grouped data (so 4 box-and whiskers in one panel)"

13. You explanation on linear regression variables is clear. Provide results for BSA in text to confirm your statement. 

17. This comment was only partially addressed. See comment 6 above.

18. The "functional state" sentence is still not clear for me. You state: "functional state after intensive care unit were found to be independent factors associated with the 6MWT distance at discharge.", while Table 4 shoes that that parameter was not significant. Please comment

22. If the authors wish a statement on postoperative activities to be added (which they did in lines 225-229), the right reference should be used. The wrong reference is now cited: REF doi:10.3390/healthcare9121735, and does not state this, but https://doi.org/10.3390/s21061979 or alternatively https://doi.org/10.3990/1.9789036550697 does.

24. Thank you. Don't forget to remove the additional dot [.] at the end of this sentence

25. It is great to see that you updated your Data Availability Statement and allow sharing of data. Unfortunately, I don't see the manuscript text changed (line 294).

Minor comments

Comments 1-3, 5-6, 8-9, 14-16, and 18-22 were answered to satisfaction, and improved the manuscript.

4. Reference 9 (in original manuscript REF 7) still focuses on late inactivity, thus insert i.e. "late" after heart surgery, line 40, to distinguish between early (your study) and late inactivity (REF 9)

7. This comment was not addressed at all

10. Thank you for the explanation. Please also change this in the manuscript, preferably in line with unadressed minor comment 7

11. Adjust manuscript accordingly

12. The authors' search to remove excessive decimals should be improved, see i.e. line 199/205/206 and throughout the manuscript, also abstract

13. This was not addressed in text. i.e. for clarity, remove "on admission"

17. Comment on the associative findings of only left extremities in the discussion.

New minor comments based on new information (Revision 1):

  1. Table 3, last row: Some text seems missing
  2. Line 225, typo: 9obilization > mobilization
  3. Line 256, insert after arbitrarily "based on the median"
  4. The reference list is inconsistently formatted, please unify
  5. Address Reviewer 4 comments, especially on methodology

General comment: please provide manuscript lines for manuscript changes to allow for easier and faster peer review.

Reviewer 3 Report

The authors have not responded to the questions asked in the first round of review. In my opinion, the article still does not meet the minimum requirements for publication. Therefore, I believe it should be rejected. 

Author Response

I would like to once again thank the reviewer for the efforts spent on evaluating our manuscript. Unfortunately, the reviewer did not see in the corrected version the answers to the comments. Nevertheless, we have done this work: added links to the introduction, finalized the discussion section and the format of the bibliography. I hope that the editors will still see our efforts on corracting the manuscript.

Reviewer 4 Report

Dear Authors, 

Thank you for making the revisions to this manuscript. I believe these revisions significantly improve the study. You have addressed all of my concerns. 

Author Response

Thank you for your help in correcting the manuscript and appreciating our work.